# NETosis and Neutrophil Activity Quantification in Pediatric Patients with Essential Thrombocythemia

**DOI:** 10.3390/ijms262411958

**Published:** 2025-12-11

**Authors:** Ekaterina-Iva A. Adamanskaya, Julia-Jessica D. Korobkin, Alexey V. Pshonkin, Alexey V. Bogdanov, Sofia V. Galkina, Nadezhda A. Podoplelova, Eugenia V. Yushkova, Mikhail A. Panteleev, Galina A. Novichkova, Nataliya S. Smetanina, Anastasia N. Sveshnikova

**Affiliations:** 1Oncology and Immunology, Dmitry Rogachev National Medical Research Center of Pediatric Hematology, 117198 Moscow, Russia; ademarrus@gmail.com (E.-I.A.A.); alexey.pshonkin@dgoi.ru (A.V.P.); alexey.bogdanov@dgoi.ru (A.V.B.); galkina.sv17@physics.msu.ru (S.V.G.); podoplelovan@yandex.ru (N.A.P.); eugenia.yushkova@gmail.com (E.V.Y.); mapanteleev@yandex.ru (M.A.P.); gnovichkova@yandex.ru (G.A.N.); nataly.smetanina@gmail.com (N.S.S.); 2Center for Theoretical Problems of Physico-Chemical Pharmacology, Russian Academy of Sciences, 109029 Moscow, Russia; juliajessika@gmail.com; 3Faculty of Physics, Lomonosov Moscow State University, 119991 Moscow, Russia

**Keywords:** neutrophil extracellular traps, myeloproliferative neoplasms, immunofluorescence microscopy, chemotaxis, flow chamber

## Abstract

Elevated levels of neutrophil extracellular traps (NETs) are associated with thrombotic risks, in particular, for patients with elevated platelet counts, such as those with essential thrombocythemia (ET). Here, the tendency for NETosis and neutrophil activity in such patients was assessed. A total of forty-one pediatric patients with elevated platelet counts diagnosed with ET (nine with CALR driver mutation, eleven with JAK2, thirteen triple-negative, and one dual-negative (TN)) or secondary thrombocytosis (five) were recruited. The tendency for NETosis was determined in a leucocyte-rich blood plasma smear using immunofluorescence staining with antibodies against myeloperoxidase and elastase. Activity of neutrophils was assessed ex vivo in parallel-plate flow chambers. The mean level of NETosis in healthy volunteers was 2.7–6.7% (95% CI). Among the ET patients, there was no statistically significant difference in NETosis level between those with mutations in CALR (19–43%), JAK2 (22–58%), and TN ones (6–27%). Patients with secondary thrombocytosis also had an elevated level of NETosis (8–66%). The velocity of neutrophil chemotaxis was significantly increased in all patients, in particular for those with mutations in CALR. These data reveal a major shift in the neutrophil activity in ET and suggest that the immunomorphological techniques presented here may allow reproducible and widely available characterization of neutrophil status.

## 1. Introduction

Thrombocytosis or elevated platelet count (higher than 450 × 10^9^/L [1]) can occur as a primary event accompanying hematological diseases, such as myeloproliferative neoplasms (MPNs), or as a secondary event such as splenectomy. Such patients are considered to have higher risks of thrombosis and thus could be treated with antiplatelet therapy [2,3]. The JAK2V617F mutation was the first discovered cause of MPN that could cause either thrombocytosis alone (essential thrombocythemia, ET) or polycythemia vera (PV), where WBC and RBC counts are also elevated [4]. Also, ET could be caused by somatic mutations in calreticulin (CALR) exon 9, MPL, or JAK2 exon 12 [5]. MPNs are associated with higher risk of thrombosis [6], with leukocytosis as a predictor of thrombosis recurrence in adults, the incidence of which decreases after starting cytoreductive therapy [7].

Recently, neutrophils and their DNA extracellular traps (NETs) were shown to contribute to thrombosis in adult MPN patients [8]. Neutrophils may increase thrombosis risks in several ways [2]. Suicidal (lytic, type I) NETosis is the canonical variant of the “explosive” death of the neutrophil [9], and it was shown to contribute highly to venous thrombosis [10,11]. This pathogenic action is manifested in diseases such as deep vein thrombosis, pulmonary embolism, and others [11,12,13]. Indeed, JAK1/2 inhibitor ruxolitinib limits NET formation and reduces venous thrombus formation [14]. Also, elevated NETosis in adult MPN patients was observed, which did not correlate with the risk of thrombosis [15]. Vital (non-lytic, type II) NETosis is characterized by DNA ejection without cell death, and it could be induced by strongly activated platelets [16] or, alternatively, attract platelets to the sites of inflammation [17]. NETosis and neutrophil activation could lead to high levels of pro-inflammatory cytokines [18] and thus favor the evolution of ET into secondary myelofibrosis [19] and cardiovascular events in MPN [20]. On the other hand, pediatric patients with MPN, especially ET, had low risks of thrombosis, but high risk of hemorrhage due to acquired von Willebrand syndrome [21]. Therefore, the neutrophil status in pediatric patients with ET is of great interest.

However, NETs do not circulate in the bloodstream and appear only upon neutrophil activation. Therefore, the level of NETosis can be observed either by means of single-cell microscopy [22], where activation occurs upon fixation, or with high-throughput methods, where artificial activation is used in solution. Gavillet et al. [23] and Masuda et al. [9] proposed a flow cytometry-based method for the quantification of NETosis level, and Singhal et al. [24] proposed an ELISA-based method. Both methods allow sufficient statistical reproducibility, but their results could not be compared with each other. Also, they do not allow distinction between the NETosis types or distinction between NETosis and other types of leukocyte death-like smudge cell formation [25].

Here, we aimed to use fluorescent microscopy of blood smears for the characterization of NETs in pediatric patients with ET and secondary thrombocytosis. We demonstrate that the EDTA-anticoagulated leukocyte-rich blood plasma smear allows optimal detection of NETs. We demonstrate that NETosis level in all patients with thrombocytosis is elevated at least two-fold, with CALR-associated ET demonstrating the highest levels of NETosis and neutrophil movement velocities.

## 2. Results

### 2.1. Neutrophils and Tendency for NETosis in LRP Smears and in the Suspension

To begin, we compared the proposed protocol for assessing the tendency toward NETosis in smears with a standard activation protocol in suspension [22,26]. Samples of leukocyte-rich plasma (LRP) from healthy donors were divided into two parts, and a smear was made from one part (see Section 4), while fixation in suspension was performed for the other part; then the sample was stained and analyzed using a fluorescent microscope (Figure 1). The nuclei of the granulocytes in suspension were deformed, with increased size, and with the “ballooning” and broadening of the “bridges” connecting the nucleus segments (Figure 1a). In the granulocytes in LRP smears, the segments in the nuclei were brightly marked and the “bridges” were thin (Figure 1c). The percentage of neutrophils undergoing NETosis in the suspension was higher than that on the smears (Figure 1e). This may be explained by using centrifugation to create a suspension of the cells.

### 2.2. Optimization of Anticoagulation

The level of spontaneous background NETosis without stimulation was compared for the LRP smears prepared from blood collected either into sodium citrate, hirudin, heparin or EDTA. Noticeably, more PMNs were observed for EDTA (Appendix A) and heparin (Appendix A) than for citrate (Appendix A), which could be explained by the PMN potentiation by citrate [27,28]. However, for heparin (Appendix A) and hirudin (Appendix A), the nuclei were deformed, which could have an extracellular calcium impact on PMNs and NETs [29]. For EDTA, the level of NETosis was significantly lower compared to other anticoagulants (Figure 1f). For the subsequent studies, EDTA was therefore used.

### 2.3. Detection of Vital and Suicidal NETosis in LRP Smears

In LRP smears, we could observe both suicidal and vital NETs (Figure 2). Vital NETs were distinguished as MPO- and elastase-positive cells without a whole nucleus, with DNA-positive granules and whole membrane observed in brightfield microscopy (Figure 2a). Suicidal NETs were distinguished by a distinct spreading of MPO- and elastase-positive DNA strands with a complete destruction of the cell membrane (Figure 2b).

To characterize and quantify the ability of neutrophils to generate NETs in response to standard NETosis-producing stimulation under conditions of the study, activation of the samples by lipopolysaccharides was performed prior to smearing. The number of NETs increased as a function of time and LPS concentration, but the total number of leukocytes was reduced, presumably since the cells were partially destroyed and could not be detected any longer (Appendix A). The LPS-induced NETosis was mostly the vital type (Figure 2c). Similarly, when PMA was used for stimulation, a small number of white blood cells remained (Appendix A). However, the suicide type of NETosis prevailed (Figure 2d).

### 2.4. NETosis and Neutrophil Activity in Essential Thrombocythemia

The level of spontaneous NETosis on the LRP smear was determined in the samples from patients with diagnosed essential thrombocythemia or secondary thrombocytosis (Table 1).

The shape of the nucleus of neutrophils and morphology of neutrophil extracellular traps in the samples were not different from those of healthy donors (Figure 3). While whole neutrophils were not different from those of healthy donors (Figure 3a,c,e,g), typical NETs were of the suicidal type and thick strands appeared in almost all of them (Figure 3b,d,f,h). Only one to two vital NETs per sample were observed for the patients; therefore, we could not reliably calculate the proportion of vital NETosis. The tendency for suicidal NETosis measured in blood smears was significantly higher (*p* < 0.001) for the patients (Figure 4a). The significance remained for each patient group based on the mutated gene as compared with the healthy controls (*p* < 0.001). We did not observe statistically significant difference between these groups, although the level of NETosis in CALR and JAK2 appeared to be greater. The therapy was not associated with a statistically significant change in the NET levels compared to the group without therapy (Figure 4a and Appendix A). Of note, patients S2 and S3 had elevated platelet counts due to iron-deficiency anemia and demonstrated the lowest and highest levels of NETosis. No correlation was found between platelet count and NETosis of any groups of patients (ρ < 0.4 in all cases, Appendix A, Table 1). For those patients who had a second visit 12 months later, the percentage of NETosis remained elevated without noticeable dynamics (Figure 4a).

Several patients from different groups who had very low levels of NETosis (patients C6, J5, J6, and T4 in Figure 4a, Table 1) were characterized by a very high level of neutrophils in the smear. This observation suggests a potential confounding relationship between neutrophil blood count and NETosis that requires further investigation. Without the listed patients, CALR and JAK2 groups demonstrate statistically significant increase in the tendency toward NETosis compared with other patients. It is also noteworthy that the patient with extremely high NETosis (J7) was the only one in the study who had hepatofibrosis, although this is but a single observation. The smears of patient J7 were characterized by the predominance of smudge cells (Appendix A). Another interesting observation was a DNA “halo” observed around the neutrophils of patients C6 and J6 (Appendix A). This may be either a patient-specific phenomenon or an artifact, and requires further investigation. For patient J6, the low allele burden could be the origin of rather low NETosis [14]; however, we did not find any associations between the allele burden and NETosis levels for other patients (Appendix A).

In order to further characterize functional activity of neutrophils in MPN, we analyzed the dynamics of their chemotactic movement in flow chambers at venous shear rate in the vicinity of growing thrombi (Figure 4b). For all MPN, as well as for secondary thrombocytosis, the median velocity of neutrophils was significantly higher than that in the control healthy donor group. The somatic mutation in CALR was associated with a greater effect. The treatment did not affect these effects greatly, although the number of samples was too low for statistical reliability. The acquired von Willebrand syndrome was associated with higher neutrophil movement velocities (Appendix A). Molecular mechanisms of this phenomenon require further investigation. The patient S5 from the secondary group with rather low neutrophil movement velocity (Figure 4b) was diagnosed with congenital sideroblastic anemia (mutation c.383A>T in ABCB7 gene) and therefore may not be representative. This indicates that MPNs are associated with a pre-activated state of neutrophils potentially caused by the excess of platelets.

## 3. Discussion

Here, we propose a novel method for determining the tendency towards NETosis in patients with MPN. By means of the proposed method, we found that, while NETosis level in healthy donors does not exceed 9%, the levels of NETosis in the pediatric patients with MPN varies from 6% for the patients without determined mutation to 60% in the patient with CALR-associated MPN (Figure 3). Additionally, we demonstrate that neutrophil chemotactic activity in MPN is significantly increased, especially in the CALR-associated MPN.

The use of the combination of blood smear with immunofluorescent techniques has recently led to a number of successes in other fields: for example, a protocol for inherited platelet disorder screening developed by Greinaher et al. was able to provide a competitive alternative to flow cytometry, not requiring expensive equipment and allowing sending the samples over long distances in order to make the method available to a wide population [30]. The method for neutrophils proposed here is based on fluorescent microscopy, which allows simultaneous visualization of DNA threads and antibody-mediated detection of neutrophil enzymes. The observed increase in the NETosis level in response to treatment with LPS or PMA (Figure 2 and Appendix A) verifies the reliability of the proposed method. The observed NET morphology also corresponds well with the results of Buhr et al. [31] and Pilsczek et al. [32]. Compared to the microscopy-based methods proposed earlier, the level of NETosis in the samples without activation is lower than in the method proposed by Fedorov et al. for NETosis detection in hematoxilin-eosin-stained blood smears [25,33], probably due to exclusion of smudge cells or another choice of anticoagulation. In other microscopy-based studies [34,35], isolation of neutrophils was used, which is known to activate PMNs [36], leading to a potential increase in background NETosis level.

Apart from the canonical suicidal NETosis, the proposed method allows observation of vital NETosis (Figure 2): neutrophils positive for MPO and elastase, with intact cell membranes, and DNA that was either missing or vesiculated. The vesiculated DNA was described earlier [16,37]. While vital NETosis was observed in response to LPS in some studies [38], we mostly observed only suicidal NETosis in these conditions, which agrees well with others [37].

NETosis is known to correlate with the higher risk of thrombosis [39], and a recent study by Wolach et al. [14] demonstrated that elevated level of NETosis is observed in JAK2-related MPN and that the level of NETosis correlates with thrombus formation in MPN. Additionally, the study of Schmidt et al. [15] also observed elevated ionomycin-induced NETosis in adult MPN. Also, the observed elevated NETosis levels and neutrophil movement velocities correspond well with previously published data for adult MPN patients [3]. Our results on elevated neutrophil velocities in ET are completely in line with the hypothesis that JAK V617F can trigger the Aim2 (absent in melanoma 2) inflammasome and lead to IL-1β generation [40,41]. While we did not detect significant differences between mutation groups or therapy groups, we found that TN patients had lower NETosis, while JAK2 group had the highest NETosis level, which is in line with the study of Schmidt et al. [15] for adults. Furthermore, we did not find significant associations between the neutrophil status and allele burden (Appendix A) or association with acquired von Willebrand syndrome (Appendix A), in line with the results of Schmidt et al. [15], which may signify a more complicated pattern of neutrophil participation in hemostasis.

The current study and the proposed method have several limitations, including, most importantly, the number of cells that can be counted in one smear. For patients with low PMN counts, the number of smears for reliable NETosis detection should be significant. Second, additional pre-analytical testing is required in order to validate the approach for interlaboratory reproducibility.

## 4. Materials and Methods

### 4.1. Reagents

Reagents include EDTA-, sodium citrate-, heparin-, and hirudin-containing S-Monovette vacuum tubes (Sarstedt, Nümbrecht, Germany), Menzel Superfrost Plus slides and coverslips (Menzel, Braunschweig, Germany), 10% goat serum (Abcam, Cambridge, UK), 2% paraformaldehyde, liquid-blocking marker (EMS), drug-pouring medium (Dako Omnis, Agilent Technologies, Santa Carla, CA, USA), LPS from *Escherichia coli* 0111:B4 (Sigma-Aldrich, St. Louis, MO, USA), phorbol-12-myristate-13-acetate (PMA), phosphate-buffered saline (PBS) (Sigma-Aldrich, St. Louis, MO, USA), Hoechst 33342 (Invitrogen, Carlsbad, CA, USA) fibrillary collagen type I (Chrono-Log Corporation, Havertown, PA, USA). Alexa Fluor^®^ 488 goat anti-mouse (A11001), Alexa Fluor^®^ 647 goat anti-rabbit (A-11011) secondary antibodies, and rhodamine phalloidin were from (Invitrogen, Carlsbad, CA, USA). The Milstein–Keller method was used to obtain mouse and rabbit anti-human monoclonal antibodies against myeloperoxidase and neutrophil elastase [42].

### 4.2. Patients and Volunteers

This is a single-center observational study. Blood from healthy donors and pediatric patients was collected at the Dmitry Rogachev Hematology Center (NMRC PHOI). The study was performed in accordance with the Declaration of Helsinki and approved by the NMRC PHOI ethics committee (protocol #8 from 18 October 2016). The group of healthy volunteers included healthy adults (ten, age 18–30) and children (seven, age 2–16). Patients were diagnosed according to the international and national guidelines based on functional and clinical analyses. The patient cohort included 34 children with ET diagnosed according WHO2016 criteria [43], two children with asplenia (one (S1) due to splenectomy for hereditary spherocytosis, and the other (S4) had asplenia as a developmental anomaly), two children (S2 and S3) with iron deficiency anemia (IDA), and one (S5) with congenital sideroblastic anemia (CSA due to hemirozygous c.383A>T p.(Tyr128Phe) p.(Y128F) (chrX:75099015 T>A) mutation *ABCB7* gene) (Table 1). Among ET patients, 9 were with CALR driver mutation, 11 with JAK2 driver mutation, 13 with triple-negative mutational status, and 1 with double-negative mutational status (patient was included to TN group). Ten patients with extremely high platelet counts (1724–2600 × 10^9^/L) had acquired von Willebrand syndrome (Table 1).

### 4.3. Immunofluorescence Microscopy of Neutrophils and NETs in LRP Smears

Venous blood drawn into different anticoagulants (EDTA K3 3 mL, lithium heparin 3 mL, 3.2% sodium citrate 3 mL or hirudin 1.6 mL) was used for the study. Leucocyte-rich plasma (LRP) was obtained by sedimentation at 37 °C for 45 min. The leukocyte count in LRP was checked to correlate well with the number of leukocytes in the complete blood count (CBC) for healthy donors (ρ > 0.8); however, for the patients, such analysis was not performed, and the CBC was performed within a week of the analysis. A small volume of LRP (3.5 μL) was applied to a positive-charged slide (SuperFrost Plus, Menzel, Braunschweig, Germany).

### 4.4. Immunofluorescent Microscopy of Neutrophils and NETs Produced in Suspension

Leukocyte-rich plasma (LRP) of heathy donors without stimulation was fixed by adding 2% formalin to LRP (4:1) for 8 min. Following this, incubation with 2% BSA (1:1) was performed for 5 min, after which the resulting mixture was centrifuged at 1500× *g* for 8 min. The supernatant was removed, and the precipitate diluted in PBS to its original volume. This mixture was then washed again at 400 g. The precipitate was diluted to a final volume of 600 μL in PBS and the suspension was applied onto glass slides coated with Poly-L. The slides were left to incubate for 15 min before being washed with mQ and air dried.

### 4.5. Fixation and Drying of Smears

Before fixation, smears were dried at room temperature for at least 24 h. They were then fixed in 2% formalin for 8 min followed by “quenching” in 2% BSA. Afterwards, the slides were washed twice in PBS.

### 4.6. Staining of Smears

LRP smears were stained using the immunofluorescent method [44]. The main steps of the proposed methodology are illustrated in Figure 5. A hydrophobic marker was used to identify the region of interest, which was then incubated with 10% goat serum for 30 min. Then, samples were incubated with primary antibodies (anti-MPO, anti-ELA) for 30 min, and washed three times in PBS. The next step was incubation with secondary antibodies (AlexaFluor 488, AlexaFluor 568) and DNA dye (Hoechst 33342). The slides were washed in PBS and the area was covered with a medium to prevent fluorescent dye burnout and coverslip. The prepared samples were analyzed using a Nikon Ti2 fluorescent microscope with confocal AX attachment (Nikon, Tokyo, Japan). PMNs were defined as cells positive for MPO and neutrophil elastase and possessing characteristic nuclear morphology (3 or 4 segments). NETosis was determined by the detection of DNA outside the cell boundaries (distinct Hoechst 33342 positive strands for suicidal NETs; distinct Hoechst 33342 positive spheres for vital NETs). For healthy donors to reliably determine NETosis, at least 100 cells were processed. Control samples from healthy donors were selected based on adherence to proper storage conditions, the absence of mechanical damage (e.g., scratches), the absence of fixation-related artifacts (such as DNA degradation), and the absence of staining artifacts (e.g., aggregation of secondary antibodies). The reliability of the method is ensured by adhering to several key pre-analytical and analytical factors: proper sample transportation conditions (maintaining a temperature of 22–37 °C, avoiding tube agitation, and processing within 4 h of blood draw), proper storage conditions for fixed samples (maintaining room temperature and humidity 40–60%), using a non-clotted sample in an anticoagulant tube, employing clean, previously unused slides, and strictly following the established protocol.

### 4.7. Canonical Induction of NETosis

NETosis was stimulated using phorbol-12-myristate-13-acetate (PMA) or lipopolysaccharides from *Escherichia coli*. The time intervals were 15 min, 30 min, or 1 h. PMA concentrations used were 50, 100, 200 or 400 nM. Lipopolysaccharide concentrations were 1, 10, or 100 ng/µL. Activators were added to LRP, which was applied to a slide after incubation.

### 4.8. Ex Vivo Fluorescent Microscopy

Parallel-plate flow chambers were described previously [45,46,47]. Channel parameters were 0.1  ×  18  ×  2 mm. Glass coverslips were coated with fibrillar collagen type I (0.2 mg/mL) for 1 h 30 min at 37 °C, washed with distilled water, and then inserted into the flow chambers. After addition of fluorescent reagents (DiOC6 (1 μM), Hoechst 33342 (2 μg/mL) and AnnexinV-Alexa647 (10 μg/mL)), blood was perfused through the parallel-plate chambers with wall shear rate of 100 s^−1^ [48]. Thrombus growth and leukocyte crawling were observed in DIC/epifluorescence modes with an upright Nikon Eclipse Ni-U microscope (Nikon, Tokyo, Japan).

For some patients, plasma dilution was performed as follows: 700 μL of blood was allowed to rest for 1 h, and afterwards 50% of plasma volume was discarded and an equal amount of Tyrode’s buffer with calcium (137 mM NaCl, 2.7 mM KCl, 12 mM NaHCO_3_, 0.36 mM NaH_2_PO_4_, 1 mM MgCl_2_, 2 mM CaCl_2_, 5 mM HEPES (pH 7.5), 0.36% BSA, 1 g/L D-glucose, pH 7.35) was added. The reasoning for dilution was as follows: for high platelet counts, the thrombus growth was so rapid that it covered most of the field of view, obscuring the determination of neutrophil trajectories (Appendix A).

### 4.9. Data Analysis

Nikon NIS-Elements software (5.22.00) was used for microscope image acquisition, and ImageJ 1.54f (http://imagej.net/ImageJ, accessed on 10 October 2023) and Python in-house scripts described earlier [49] were utilized. Tracking code listing and program operation examples are available at (https://github.com/juliajessika/Leukocytes2023, accessed on 24 May 2024). Statistical analysis was performed using Python 3.8 and GraphPad Prism 8 software (GraphPad Software, La Jolla, CA, USA) using Mann–Whitney U test, and the significance level was set as 95%. Where appropriate, Spearman’s correlation coefficient (ρ) was calculated in the GraphPad Prism 8 software.

## Figures and Tables

**Figure 1 ijms-26-11958-f001:**
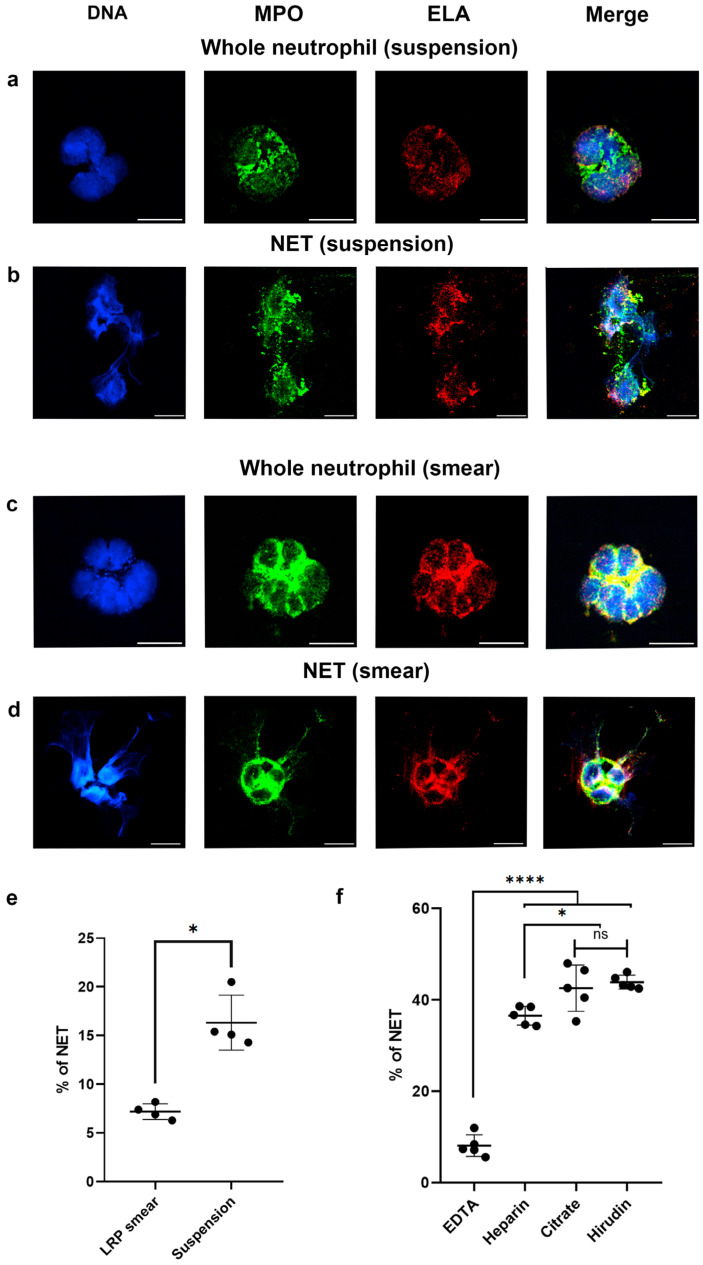
Comparative cell morphology and statistics of neutrophils and NETs observed in cell suspension and in LRP smears from healthy donor samples. (**a**,**b**) Representative micrographs of cells from suspension. (**a**) Whole neutrophil. (**b**) NET. (**c**,**d**) Representative micrographs of cells in LRP smear (EDTA). (**c**) Whole neutrophil. (**d**) NET. (**a**–**d**) Blue, DNA (Hoechst 33342); green, MPO; red, ELA (neutrophil elastase); scale bar 10 μm. (**e**) Levels of NETosis when PFA-fixation was performed in cell suspension or in LRP smear (*n* = 4). (**f**) Level of NETosis in LRP smear produced from blood samples collected into different anticoagulants (*n* = 5). Mann–Whitney U test was used to compare two independent samples. ns, non-significant; * corresponds to *p* < 0.05; **** corresponds to *p* < 0.0001.

**Figure 2 ijms-26-11958-f002:**
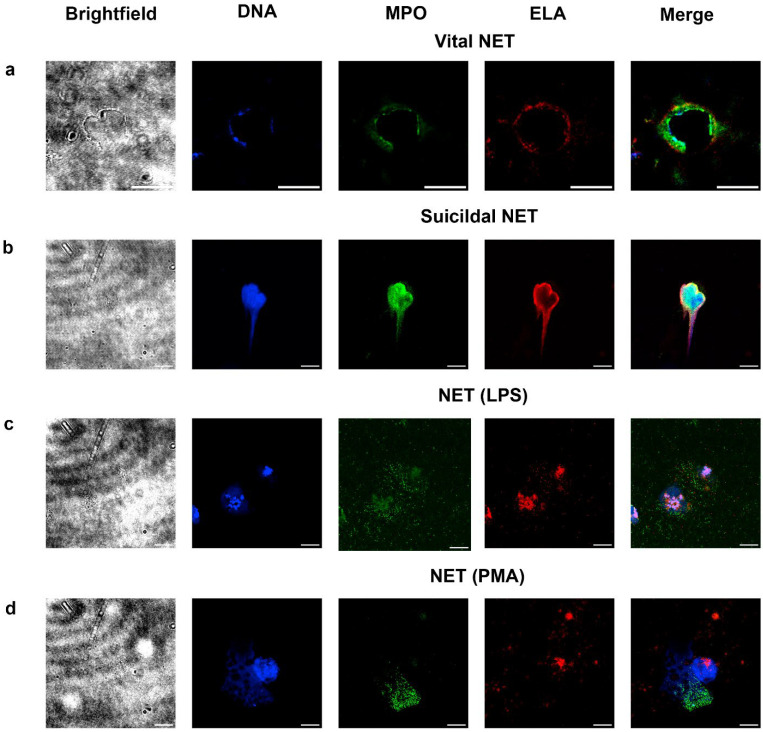
Scenarios of NETosis observed in LRP smears. (**a**) Typical vital NETosis in a sample without stimulation. (**b**) Typical suicidal NETosis without stimulation. (**c**) Typical NET, stimulation with LPS. (**d**) Typical NET, stimulation with PMA. Blue, DNA (Hoechst 33342); green, MPO; red is ELA (neutrophil elastase); scale bar is 10 μm. Only for vital NETosis, some part of cell membrane could be seen in brightfield. For suicidal NETs, nothing could be seen.

**Figure 3 ijms-26-11958-f003:**
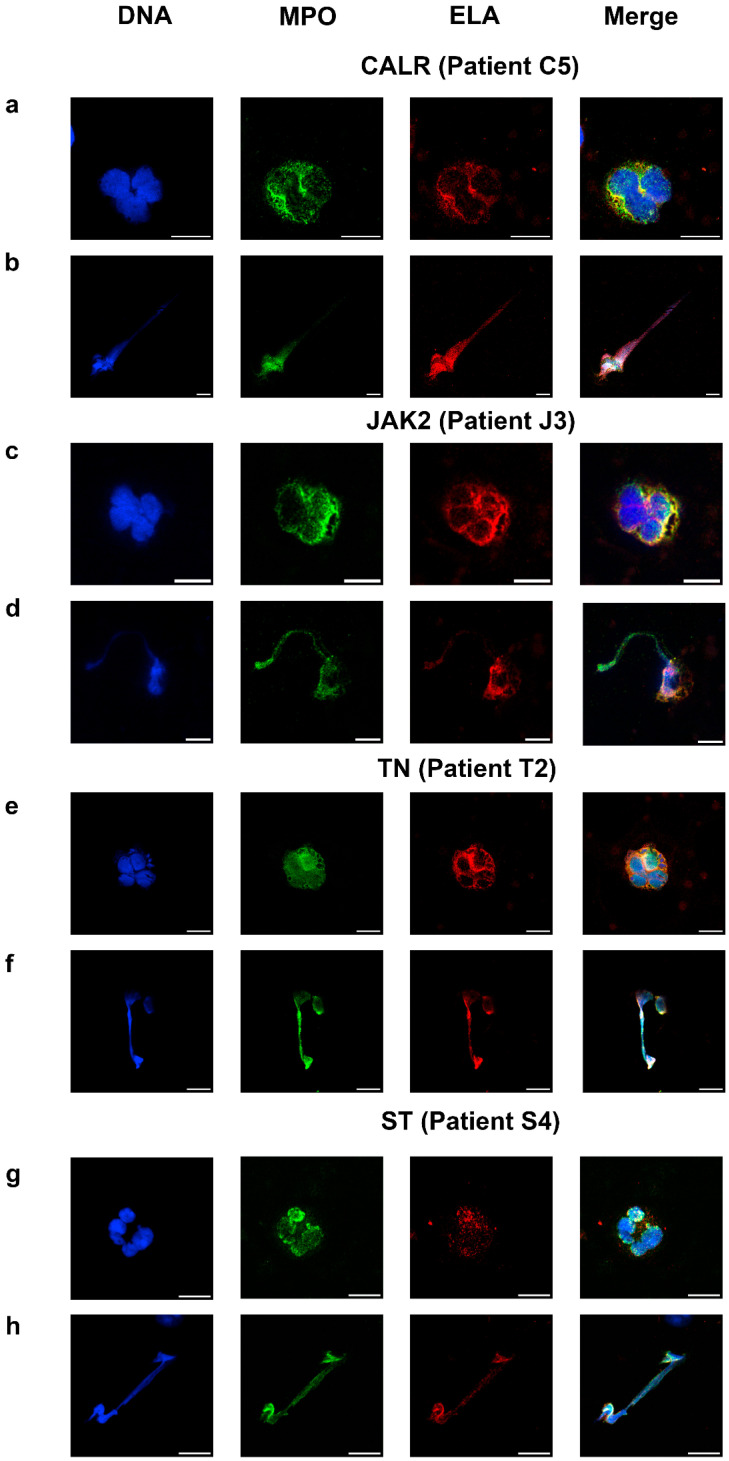
Cell morphology and quantity of NETs in the MPN patients. (**a**,**b**) WN and NET of a patient with a mutation in exon 9 of the CALR gene; (**c**,**d**) WN and NET of a patient with a mutation in exon 14 of the JAK2 gene; (**e**,**f**) WN and NET of patient lacking demonstrable mutations affecting JAK2, CALR, or MPL driver genes (TN, triple negative). (**g**,**h**) WN and NET of patients with secondary thrombocytosis (ST). Blue, DNA (Hoechst 33342); green, MPO; red is ELA (neutrophil elastase); scale bar 10 μm.

**Figure 4 ijms-26-11958-f004:**
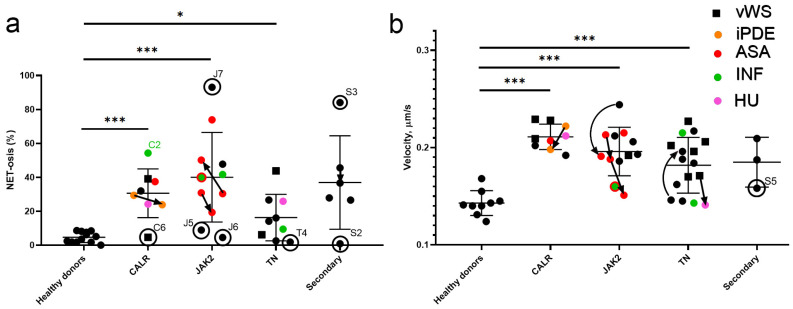
Quantification of neutrophil activity in the patients with myeloproliferative neoplasms. (**a**) NETosis on the smear; (**b**) velocity of chemotaxis in flow chambers. The figures show circles for individual patients, as well as medians and quartiles for the groups: healthy donors, essential thrombocytemia (ET) with CALR driver mutation (CALR), ET with JAK2 driver mutation (JAK2), ET with triple-negative (TN) mutational status, and secondary thrombocytosis. The shape and color indicate additional diagnoses and therapy of individual patients: squares, acquired von Willebrand syndrome (vWS); orange, anagrelide (iPDE); red, acetylsalicylic acid (ASA); green, pegylated interferon alfa-2a (INF); pink, hydroxyurea (HU). Arrow connections show the data for the same patient on consecutive visits, from the first visit to the second one. Circles indicate patients discussed in the text and named in the Table 1. Mann–Whitney U test was used to compare two independent samples; * corresponds to *p* < 0.05, *** corresponds to *p* < 0.001.

**Figure 5 ijms-26-11958-f005:**
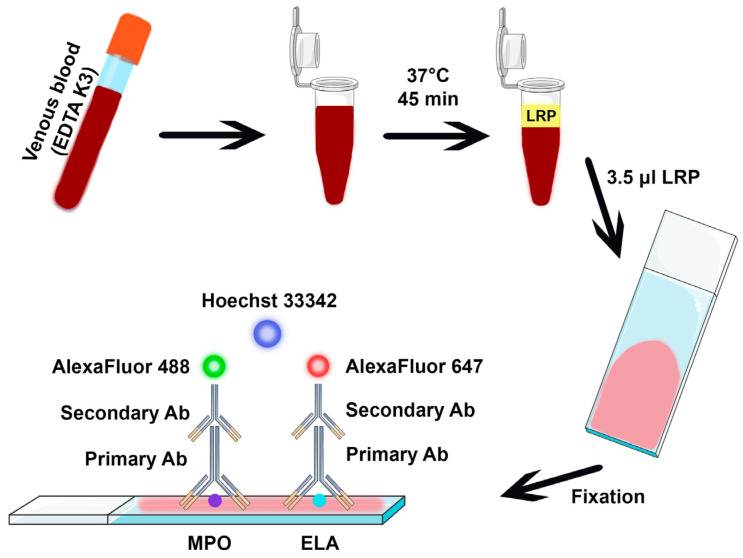
Schematic representation of the proposed method. A total of 1 mL from the anticoagulated venous blood sample (EDTA K3 tubes) is left for sedimentation to obtain leukocyte-rich plasma (LRP), then the LRP smear is prepared on a positively-charged slide, and after drying in the air, the slide is fixed by formalin solution, washed, and then stained with fluorescently labeled antibodies for myeloperoxidase (MPO) and neutrophil elastase (ELA).

**Table 1 ijms-26-11958-t001:** Patients with elevated platelet counts included in the study. CALR, JAK2, and TN—subgroups among ET patients. Secondary is the group of patients with secondary thrombocytosis due to asplenia, IDA, or CSA.

Group	Patient	Genetics	Point	Age, y	PLT ^##^ 204–356 10^9^/L	NEUT ^##^ 1.5–8.5 10^9^/L	Acquired von Willebrand Syndrome	Treatment *	NET/PMNs
CALR	C1	CALR (exon 9) 42.8% ^$^	1	12	617	5.44		iPDE	63/214
			2	12	230	1.67		iPDE	11/46
	C2	CALR (exon 9)	1	13	393	2.7		INF	89/164
	C3	CALR (exon 9) 37% ^$^	1	15	894	6.19		-	8/25
	C4	CALR (exon 9)	1	17	1695	5.03		ASA	54/144
	C5	CALR (exon 9) 11% ^$#^	1	15	167	1.92		HU	19/78
	C6	CALR (exon 9) 31% ^$^	1	9	1868	7.09	+	-	10/213
	C7	CALR (exon 9) 39.5% ^$^	1	7	1832	5.48	+	-	9/23
	C8	CALR (exon 9) 59% ^$^	1	6	865	3.07		-	-
	C9	CALR (exon 9)	1	8	2406	7.94	+	-	-
JAK2	J1	JAK2 (V617F)	1	17	310	1.88		INF	10/24
	J2	JAK2 (V617F) 26% ^$^	1	13	404	1.46		ASA, INF	10/25
	J3	JAK2 (V617F)	1	10	1651	5.34		ASA	24/79
			2	10	1351	6.83		ASA	-
			3	11	678	4.75		ASA	100/199
	J4	JAK2 (V617F) 6% ^$^	1	10	1199	4.61		ASA	52/168
			2	11	1337	5.13		ASA	74/381
	J5	JAK2 (V617F) 11% ^$#^	1	16	589	3.96		-	30/332
	J6	JAK2 (V617F) 5.5% ^$^	1	16	830	7.54		-	4/87
	J7	JAK2 (V617F)	1	4	1280	7.7		-	3/44 ^&^
	J8	JAK2 (V617F) 15% ^$#^	1	17	1156	7.71		-	44/92
	J9	JAK2 (V617F) 14% ^$^	1	16	1151	10.06		-	-
			2	17	1196	10.3		ASA	99/134
	J10	JAK2 (V617F) 39% ^$^	1	5	1172	8.34		-	-
	J11	JAK2 (V617F) 32% ^$#^	1	16	1724	7.61	+	-	-
TN	T1	Negative: CALR, CALR-indel, MPL, JAK2(12,14)	1	11	943	4.42		-	44/165
	T2	Negative: CALR, CALR-indel, MPL, JAK2(12,14)	1	17	495	1.52		INF	15/157
	T3	Negative: CALR, CALR-indel, MPL, JAK2(12,14) ^#^	1	13	1701	6.32	+	-	32/73
	T4	Negative: MPL, JAK2	1	9	811	4.13		-	7/366
	T5	Negative: CALR, MPL, JAK2(12,14), BCR\ABL1	1	11	1840	9.63	+	-	12/192
			2	12	439	1.89		HU	7/27
	T6	Negative: CALR, MPL, JAK2(12)	1	14	1072	3.94		-	27/168
	T7	Negative: CALR, MPL, JAK2(12)	1	14	657	2.15		-	-
			2	15	848	3.4		-	4/152
	T8	Negative: CALR, CALR-indel, MPL, JAK2(12,14), 515L	1	4	810	4.41		-	3/21
	T9	Negative: CALR, CALR-indel, MPL, JAK2(12,14)	1	10	918	4.63		INF	-
	T10	Negative: CALR, CALR-indel, MPL, JAK2(12,14)	1	3	2600	4.68	+	-	-
	T11	Negative: CALR, CALR-indel, MPL, JAK2(12,14)	1	13	1323	4.5		-	-
	T12	Negative: CALR, MPL, JAK2(12,14)	1	7	1903	4.31	+	-	-
	T13	Negative: CALR, CALR-indel, MPL, JAK2(12,14)	1	6	1964	4.22	+	-	-
	T14	Negative: CALR, CALR-indel, MPL, JAK2(12,14)	1	16	1824	6.85	+	ASA	-
Secondary	S1	Hereditary spherocytosis after splenectomy	1	6	762	3.94		-	36/79
			2	7	976	2.43		-	18/49
	S2	IDA	1	6	602	3.17		-	2/214
	S3	IDA	1	11	460	1.44		-	37/44
	S4	Asplenia	1	7	827	6		-	131/469
	S5	Congenital sideroblastic anemia (*ABCB7* c.383A>T hetero)	1	14	535	3.55		-	73/274

* Treatment: INF, pegylated interferon alfa-2a; ASA, acetylsalicylic acid; HU, hydroxyurea; iPDE, anagrelide; ^#^ IDA, iron-deficiency anemia; ^&^ smudge cells dominate; ^$^ Allele burden; ^##^ complete blood cell count (CBC).

## Data Availability

The original contributions presented in this study are included in the article/Appendix A. Further inquiries can be directed to the corresponding author.

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
