# Peer review of "NETosis and Neutrophil Activity Quantification in Pediatric Patients with Essential Thrombocythemia"

_ijms, 2025, doi:10.3390/ijms262411958_

Round 1
Reviewer 1 Report
Comments and Suggestions for Authors
The authors developed a novel method for NETosis level determination by use of blood smear of leukocyte rich plasma and immunofluorescence. This method was used to discover a significant increase of NETosis in MPN patients.
Comments:
- What might be the reason for the lower percentage of NETosis in EDTA anticoagulated blood compared with other anticoagulants?
- A clear description for NETs determination should be given in the method section. How many cells were assessed per blood smear?
- Since NETs were formed upon fixation on smear, not directly from blood circulation, I believe the observed results should be termed higher NETosis tendency, to be more accurate..
- Although the authors described that NETosis in MPN blood were mostly of the suicide type, I wonder how high is the percentage for vital NETosis
- Typos that should be corrected, “MNP” at line 57; “patient count” at line 141.
Author Response
The authors developed a novel method for NETosis level determination by use of blood smear of leukocyte rich plasma and immunofluorescence. This method was used to discover a significant increase of NETosis in MPN patients.
Thank you for your kind comments on our manuscript. We re-arranged the manuscript and provided additional schemes in order to cover the issues that you have found in our work.
- What might be the reason for the lower percentage of NETosis in EDTA anticoagulated blood compared with other anticoagulants?
Thank you for this comment! We expect this phenomenon to be related to the fact that citrate and heparin potentiate neutrophils for activation. We have added the following sentence to the Results:
“Noticeably more PMNs were observed for EDTA (Fig. S2A,B) and heparin (Fig. S2D) than for citrate (Fig. S2C), which could be explained by the PMN potentiation by citrate [25,26]. However, for heparin (Fig. S2E) and hirudin (Fig. S2F), the nuclei were deformed, which could by the extracellular calcium impact on PMNs and NETs [27].”
- A clear description for NETs determination should be given in the method section. How many cells were assessed per blood smear?
Thank you for noticing this oversight on our part! We have added the corresponding description into the Methods section: “PMNs were defined as cells positive for MPO and neutrophil elastase and possessing characteristic nuclear morphology (3 or 4 segments). NETosis was determined by the detection of DNA outside the cell boundaries (distinct Hoechst 33342 positive strands for suicidal NETs; distinct Hoechst 33342 positive spheres for vital NETs). For healthy donors to reliably determine NETosis, at least 100 cells were processed.
The number of cells observed for each patient is given in Table 1.
- Since NETs were formed upon fixation on smear, not directly from blood circulation, I believe the observed results should be termed higher NETosis tendency, to be more accurate.
We thank you for this comment! We fully agree with the reviewer's opinion; however, all existing microscopic or flow cytometry methods also use fixation and, therefore, tend to observe a tendency toward NETosis rather than NETosis, yet they use the term "NETosis." Therefore, we have not changed the article title, but have corrected the terminology in the text.
- Although the authors described that NETosis in MPN blood were mostly of the suicide type, I wonder how high is the percentage for vital NETosis
Thank you for this suggestion! Unfortunately in our analysis only single vital NETs were found. We have added the corresponding sentence to the Results section: “Only 1-2 vital NETs per sample were observed for the patients, therefore, we could not reliably calculate the proportion of vital NETosis.”
- Typos that should be corrected, “MNP” at line 57; “patient count” at line 141
Corrected, thank you!

Reviewer 2 Report
Comments and Suggestions for Authors
The authors conducted a study to investigate NETosis and neutrophil activation in pediatric patients with essential thrombocythemia. The question raised is interesting, and the results are well documented, but several issues need clarification before the paper can be accepted. My comments are listed below:
• The introduction is clear and provides sufficient information regarding the subject of the article.
• Although the list of abbreviations can be found at the end of the manuscript, they should be used in the text where they first appear. Please check the entire text, as in many places either the abbreviation or the explanation is missing when first used, e.g., Line 24: TN; Line 77: LRP smears
• 14 triple-negative cases were analyzed, but according to Table 1, patient T4 is only double-negative. Please check the data in the table.
• In line 56, should "MNP" be MPN?
• In Figure 1c, the magnification should be the same as in part a for better comparability. Similarly, the magnification in parts b and d should also be the same.
• In Figure 1e, the labels on the x-axis should be further apart.
• Results on "Neutrophils in LRP smears and in the suspension": What samples were used in this experiment?
• Use the term "LRP smear" consistently throughout the manuscript instead of "smear samples" or "plasma smear."
• What were the selection and exclusion criteria for the control samples?
• There are some inaccurate statements and incomplete data in the Materials and Methods section:
Line 269: "A small volume of LRP was applied to an adhesive-coated slide." Please specify the volume and the type of adhesive-coated slide.
Line 293: "Activators were added to blood plasma, which was applied to a slide after incubation." Blood plasma or RLP?
Line 297: The source of collagen is missing.
Data analysis: The calculation of the correlation coefficient is not included in the statistical analyses.
• In Table 1: Are the PLT and NEUT numbers measured in whole blood or leukocyte-rich plasma? Was it examined whether the neutrophil count in LRP plasma changed after sedimentation compared to whole blood? Using the term "point" instead of the sample number is confusing. What did the NET/PMNs value in the last column of the table depend on? It does not seem to correlate with the original neutrophil count.
• Lines 123-125: Spontaneous NETosis was examined not only in patients with ET, but also in control samples.
• Lines 139-140: "The therapy was not associated with statistically significant changes in the NET levels compared to the group without therapy (Fig. 4a)." Figure 4a does not show such a comparison.
• Lines 142-143: "No correlation was found between platelet count and NETosis of any groups of patients (Rs < 0.4 in all cases, Table 1)." Please provide the results of the correlation tests; even an r < 0.4 result can be interpreted if the association is statistically significant.
• Line 167: "The patient S7 from the secondary group." There is no such case in the table. S5?
• Lines 204-205: "and detection of active neutrophil enzymes"; how was the activity of the enzymes verified? They were detected as antigens using antibodies.
• It would be useful for readers if a diagram were provided showing the main steps/parameters of the proposed method.
• Please summarize the important preanalytical and analytical factors that ensure the reliability of the method.
Author Response
Reviewer #2:
The authors conducted a study to investigate NETosis and neutrophil activation in pediatric patients with essential thrombocythemia. The question raised is interesting, and the results are well documented, but several issues need clarification before the paper can be accepted.
We are grateful for your propositions! We re-arranged the manuscript and provided additional schemes in order to cover the issues that you have found in our work.
- The introduction is clear and provides sufficient information regarding the subject of the article. Although the list of abbreviations can be found at the end of the manuscript, they should be used in the text where they first appear. Please check the entire text, as in many places either the abbreviation or the explanation is missing when first used, e.g., Line 24: TN; Line 77: LRP smears
Thank you for pointing out these shortcomings, we have fixed everything!
- 14 triple-negative cases were analyzed, but according to Table 1, patient T4 is only double-negative. Please check the data in the table.
We are grateful to the Reviewer for pointing out this shortcoming! We have corrected the patient descriptions to avoid misleading readers.
In Abstract: “41 pediatric patients with elevated platelet counts diagnosed with ET (9 with CALR driver mutation, 11 with JAK2, and 13 triple-negative and one dual-negative (TN)) or secondary thrombocytosis (5) were recruited.”
In Methods: “Among ET patients, 9 were with CALR driver mutation, 11 with JAK2 driver mutation and 13 with triple-negative mutational status, 1 with double-negative mutational status (patient was included to TN group).”
- In line 56, should "MNP" be MPN?
Yes, corrected.
- In Figure 1c, the magnification should be the same as in part a for better comparability. Similarly, the magnification in parts b and d should also be the same.
Thank you for pointing this out, we have corrected this issue!
- In Figure 1e, the labels on the x-axis should be further apart.
Thank you for pointing this out, we have corrected this issue!
- Results on "Neutrophils in LRP smears and in the suspension": What samples were used in this experiment?
Thank you for pointing this out! We have modified the first paragraph of the Results for clarification: “To begin, we compared the proposed protocol for assessing the tendency toward NETosis in smears with a standard activation protocol in suspension [17]. Samples of leukocyte rich plasma (LRP) from healthy donors were divided into two parts, then a smear was made from one part (see Methods), and fixation in suspension was performed for the other part, then the sample was stained, and analyzed using a fluorescent microscope (Fig. 1).”
- Use the term "LRP smear" consistently throughout the manuscript instead of "smear samples" or "plasma smear."
Corrected
- What were the selection and exclusion criteria for the control samples?
Corrected: “Control samples from healthy donors were selected based on adherence to proper storage conditions, the absence of mechanical damage (e.g., scratches), the absence of fixation-related artifacts (such as DNA degradation), and the absence of staining artifacts (e.g., aggregation of secondary antibodies).”
- There are some inaccurate statements and incomplete data in the Materials and Methods section:
- Line 269: "A small volume of LRP was applied to an adhesive-coated slide." Please specify the volume and the type of adhesive-coated slide.
Corrected: “A small volume of LRP (3.5 μl) was applied to a positive charged slide (SuperFrost Plus, Menzel, Braunschweig, Germany)”.
- Line 293: "Activators were added to blood plasma, which was applied to a slide after incubation." Blood plasma or RLP?
Corrected: “Activators were added to LRP, which was applied to a slide after incubation”
- Line 297: The source of collagen is missing.
Corrected: “… fibrillary collagen type I (Chrono-Log Corporation; Havertown; USA)”.
- Data analysis: The calculation of the correlation coefficient is not included in the statistical analyses.
Corrected: “Where appropriate, Spearman’s correlation coefficient (ρ) was calculated in GraphPad Prism 8 software.”
- In Table 1: Are the PLT and NEUT numbers measured in whole blood or leukocyte-rich plasma? Was it examined whether the neutrophil count in LRP plasma changed after sedimentation compared to whole blood? Using the term "point" instead of the sample number is confusing. What did the NET/PMNs value in the last column of the table depend on? It does not seem to correlate with the original neutrophil count.
Thank you for this comment! The neutrophils were calculated during a routine CBC test, which was not necessary taken on the same day as the analysis. We have included the following statement in the Methods section: “The leukocyte count in LRP was checked to correlate well with the number of leukocytes in the complete blood count (CBC) for healthy donors (ρ > 0.8), however, for the patients such analysis was not performed, and the CBC was performed within a week of the analysis.”
We have corrected each usage of the word “point”.
- Lines 123-125: Spontaneous NETosis was examined not only in patients with ET, but also in control samples.
Corrected.
- Lines 139-140: "The therapy was not associated with statistically significant changes in the NET levels compared to the group without therapy (Fig. 4a)." Figure 4a does not show such a comparison.
Corrected: new Figure S6 demonstrates the comparison.
- Lines 142-143: "No correlation was found between platelet count and NETosis of any groups of patients (Rs < 0.4 in all cases, Table 1)." Please provide the results of the correlation tests; even an r < 0.4 result can be interpreted if the association is statistically significant.
Corrected: new Figure S6b demonstrates the absence of correlation.
- Line 167: "The patient S7 from the secondary group." There is no such case in the table. S5?
Corrected!
- Lines 204-205: "and detection of active neutrophil enzymes"; how was the activity of the enzymes verified? They were detected as antigens using antibodies.
Thank you for pointing this out! We have not measured the enzyme activity, the corresponding sentence was corrected.
- It would be useful for readers if a diagram were provided showing the main steps/parameters of the proposed method.
Corrected. New Figure 5 illustrates the main steps of the process
- Please summarize the important preanalytical and analytical factors that ensure the reliability of the method.
The following paragraph was added in the Methods: “The reliability of the method is ensured by adhering to several key preanalytical and analytical factors: proper sample transportation conditions (maintaining a temperature of 22–37°C, avoiding tube agitation, and processing within 4 hours of blood draw), proper storage conditions for fixed samples (maintaining room temperature and humidity 40-60%), using a non-clotted sample in an anticoagulant tube, employing clean, previously unused slides, and strictly following the established protocol.”

Round 2
Reviewer 2 Report
Comments and Suggestions for Authors
I have no further comments.